# Verification of Particle Tracking and In Situ Tracer Experiment for the Gravel and Cholan Formation Composite in Northwest Taiwan

**Cong-Zhang Tong [1,*], Pin Yeh [2], Yun-Chen Yu [1], Liang-Gu Chen [3] and Han-Hsiang Tseng [3]**

[1] Department of Chemical Engineering, National Atomic Research Institute, Taoyuan City 325207, Taiwan; yuyc@nari.org.tw
[2] Fuel Cycle and Materials Regulation Office, Nuclear Safety Commission, New Taipei City 234, Taiwan; chrissonic2000@gmail.com
[3] Department of Radiation Protection, National Atomic Research Institute, Taoyuan City 325207, Taiwan; rliang@nari.org.tw (L.-G.C.); hansiang@nari.org.tw (H.-H.T.)
[*] Correspondence: cztong@nari.org.tw; Tel.: +886-3-4711400 (ext. 5638)

**Abstract:** This paper presents the verification results of an experimental site that employed a particle tracking algorithm to assess the transport of tracers through the composite formation of gravel and Cholan in northwest Taiwan. A suitable hydrogeological conceptual model that describes the flow characteristics of the gravel formation and Cholan formation is essential to evaluate groundwater flow and management at this site. Continuous porous medium (CPM) can be easily applied in the gravel formation, while the Cholan formation, characterized by argillaceous sandstone, is commonly treated as a porous medium. However, this study obtains its fracture distribution through geological surveys, and the key fracture parameters are also collected, analyzed, and incorporated into the model. Four hydrogeological conceptual models, including CPM, discrete fracture network (DFN), equivalent continuous porous medium (ECPM), and hybrid DFN/ECPM, are generated for this complex formation. This study combines the conceptual models of the gravel and Cholan formation into four cases to describe the characteristics of the composite formation. The groundwater flow field of four cases is simulated, and the particle tracking method is employed to model the tracer transport. Simulation results from the four hybrid models all yielded a breakthrough curve (BTC) for the first 15 h, indicating that the tracer arrived at the designated outlet within this timeframe and primarily flowed through the gravel formation, while long-time particle tracking revealed a possible flow path through the Cholan formation after 15 h. The breakthrough curve of the four cases shows that the ECPM model is more suitable for representing the heterogeneity of the Cholan formation than the common CPM model. This study provides a suitable numerical algorithm of the conceptual model of the Cholan formation based on strong evidence by considering different models and comparing them with in situ tracer tests.

**Keywords:** gravel formation; Cholan formation; groundwater flow; particle tracking; tracer test

## 1. Introduction

The in situ tracer test is a classical investigation technique used to identify and characterize groundwater flow and transport mechanisms in aquifers. The tracer test involves tracking groundwater flow in geological formations using natural or man-made chemicals [1–3]. Geologists and hydrogeologists routinely leverage the feedback from in situ tracer tests to understand the connectivity of various mediums within the geological formation, evaluate the flow direction and velocity of groundwater, confirm watershed boundaries, and formulate follow-up monitoring plans [4]. Tracers commonly employed include saline solutions, fluorescent agents, radioisotopes, bromides, etc. Each tracer exhibits a different performance in tests based on its physical and chemical properties. However,

when a complete concentration–time curve can be derived from tests, these tracers can be applied to analyze the hydraulic properties of research sites [5].

Among these tracers, bromide is considered one of the best hydrological tracers due to its conservative behavior in most environments. Bromide ions naturally occur in the environment as halides in the ocean, alleviating concerns about environmental pollution and threats to human health when using sodium bromide as a tracer. Furthermore, sodium bromide is cost-effective and straightforward to measure [6,7]. Bromide ions are less prone to chemical reactions with the matrix because they are repelled by most negatively charged matrices and tend to concentrate in the center of the water molecule [8]. In most tests using bromide ions as tracers, the impact of chemical reactions and adsorption between solutes and substrates on the transport mechanism can be assumed to be minimal or even negligible. The timescale and spatial scale are critical considerations when evaluating solute transport in geological formation using in situ tracer tests. These tests typically operate on relatively short timescales, in conjunction with the retention and adsorption capacity of the geological formation. As a result, experiments are constrained to limited spatial scales. Long-term tracer tests covering distances of more than a few kilometers typically extend over several years to a maximum of ten years, while short-term tracer tests, covering distances of only a few hundred meters, may span hours to weeks or months [9].

Groundwater flow and transport simulations are a common approach to modeling the flow and solute transport mechanisms underground. Numerical methods combined with grids, such as the finite difference method (FDM), the finite volume method (FVM), and the finite element method (FEM), have been widely applied to the numerical calculation of groundwater flow and solute transport. Different numerical methods may cause minor discrepancies between flow fields [10]. The algorithms used for transport simulations are another factor that can influence the results. Among all the transport models, the particle tracking method is a typical approach to simulate advective transport in geological media [10–12]. Traditional particle tracking methods and random walk particle tracking methods are commonly used to simulate solute transport regardless of whether it is in the FDM, FVM, or FEM. Additionally, Smoothed Particle Hydrodynamics (SPH) is another novel method, which is a mesh-free particle method based on the Lagrangian formulation [13]. Herrera et al. [14] derived a meshless numerical method based on SPH for the simulation of conservative solute transport in heterogeneous geological formations. SPH offers significant advantages over traditional grid-based numerical models in handling large deformations; tracking free surfaces, moving interfaces, and deformable boundaries; and resolving moving discontinuities such as the interface between fractures and rocks [15]. SPH holds great potential for addressing many problems in engineering and science. The aforementioned studies employ numerical methods to solve groundwater flow and transport problems; however, all simulated results should be validated for accuracy.

An integrated approach that combines a hydrogeological conceptual model with in situ tracer tests will facilitate the mutual validation of numerical simulations and in situ test results. This comprehensive approach can provide valuable insights to decision-makers involved in resource management and planning. In alignment with the geological characteristics of the experimental site, this study constructs various hydrogeological models and models the associated groundwater flow and transport using FracMan software (version 7.9). It simulates the travel length, travel time, and concentration of particles from the upstream release well to the downstream observation position using the particle tracking algorithm. By comparing the simulation results of particle tracking with the outcomes of in situ tracer experiments, this study explores the correlation between numerical simulations and in situ tracer experiments, ultimately establishing the appropriate geological model for this specific experimental site.

The site of this study is located in a slope land scheduled for the projects of slope improvement and site strengthening of the composite formation of gravel and Cholan in northwest Taiwan. In this project, the characteristics of the groundwater flow at the site, such as flow velocity and direction, are important factors that affect the effectiveness of the

project. A series of preparatory works have been carried out for this purpose, including geological surveys, material property tests, tracer tests, numerical simulations, etc. This study is a summary of the preparatory works for this project. It is hoped that the optimal description model for groundwater flow and long-term solute transport trends at the site can serve as a valuable reference for the project. It can also establish a connection between civil engineering and geological engineering. Additionally, it aims to provide a suitable numerical algorithm for the conceptual model of the Cholan formation, based on strong evidence derived from considering various models and comparing them with in situ tracer tests.

## 2. Methods

### 2.1. Continuum and Discrete Approach

Determining the appropriate hydrogeological conceptual model is a crucial initial step when simulating groundwater flow and transport in an aquifer or formation. For most sedimentary formations, it is challenging to determine whether they consist primarily of fractures embedded in the rock matrix or are predominantly porous medium. This complexity arises from significant differences in hydraulic and transport properties due to the high heterogeneity between fractures and the surrounding matrix. In recent years, algorithms for both continuous and discrete approaches have been developed to investigate, understand, and ultimately predict flow and transport behavior in such complex systems. The selection of an appropriate conceptual model depends on the spatial scale of the region of interest and the characteristics of the fractured aquifer system.

The continuum approach treats a fractured aquifer system as homogeneous and continuous, with its hydraulic properties averaged and assumed to be constant over the domain. Such a fractured aquifer can be described as a continuous porous medium (CPM) model [16–18]. The fractured aquifer is represented by the lumped hydraulic parameters, and the flow and transport processes can be described using continuity equations. The advantage of the CPM model lies in its simplicity. It disregards the complex geometry of fractured systems and instead introduces averaged hydraulic parameters (such as hydraulic conductivity, porosity, etc.) to describe the flow and transport behavior. These hydraulic parameters can be obtained from field measurements [19]. However, the flow and behavior of fractured systems may resemble those of a homogeneous porous medium when the size of the study area increases, fracture density increases, and fracture orientation changes rather than remaining constant [20]. In terms of the spatial scale of numerical simulation, the CPM model has proven to be a promising method for large fields ranging from hundreds of meters to several kilometers, and it can be used to simulate regional groundwater flow fields at large scales, such as the Yucca Mountains [21]. In terms of rock properties, CPM models are considered appropriate for certain types of rock and materials in which flow is predominantly through an inter-connected network of pores in the rock matrix, such as for many clastic rocks, or for soils and unconsolidated deposits [22].

The discrete approach can unambiguously determine the spatial location, geometric dimensions, and hydraulic properties of fractures. Detailed investigation and geostatistical methods are required to obtain fracture geometry, including orientation, size, aperture, and density. The discrete fracture network (DFN) model is a discrete approach that can independently study the effect of fracture geometry on hydraulic behavior. It can represent preferential flow paths arising from high transmissivity in connected fractures. The DFN model is widely applicable to fractured rocks, where groundwater flow is more likely to flow within highly conductive connecting fractures than in an impermeable rock matrix. Studies have shown that the DFN model is employed to simulate groundwater flow, and results indicate that the simulation results of the random DFN model are in good agreement with the observation data [23,24].

Oda estimated the permeability tensor after generating a two-dimensional DFN model based on parameter values and distribution patterns such as fracture position, length, and aperture [25]. The permeability coefficient of each fracture can be averaged to obtain the

equivalent permeability coefficient of the rock matrix. This is because the continuum mode lumps the hydraulic properties of fractures and the rock matrix together, and is referred to as the equivalent continuous porous media (ECPM) model. The hydraulic properties of each cell are represented by the equivalent hydraulic parameters obtained from the Oda upscaling analysis. The ECPM model is suitable for large-scale and well-permeable rock matrices.

The permeability tensor in three dimensions for each cell can be calculated through Oda upscaling analysis, known as Oda tensor, which serves as a simplification of Darcy's law for laminar flow through an isotropic porous medium. For a specific cell with a known fracture area ($A_k$) and transmissivity ($T_k$) from the DFN model, the fracture tensor can be calculated by summing the contributions of individual fractures, each weighted by its area and transmissivity:

$$F_{ij} = \frac{1}{V} \sum_{k=1}^{N} A_k T_{f,k} n_{ik} n_{jk} \tag{1}$$

where $F_{ij}$ is the fracture tensor (L/T), i and j are the two perpendicular directions on the fracture plane, V is the cell volume ($L^3$), N is the total number of fractures in each cell (-), $A_k$ is the area of fracture k ($L^2$), $T_{f,k}$ is the transmissivity of fracture k ($L^2/T$), and $n_{ik}$ and $n_{jk}$ are the components of a unit normal to the fracture k (-).

Oda's permeability tensor is derived from $F_{ij}$ by assuming that $F_{ij}$ expresses fracture flow as a vector along the fracture's unit normal:

$$k_{ij} = \frac{1}{12} \left( F_{kk} \delta_{ij} - F_{ij} \right) \tag{2}$$

where $k_{ij}$ is a permeability tensor (L/T), $F_{kk}$ and $F_{ij}$ are components of the fracture tensor (L/T), and $\delta_{ij}$ is Kroenecker's delta (-). If i = j, then $\delta_{ij}$ = 1; otherwise, $\delta_{ij}$ = 0.

## 2.2. Hydrogeological Conceptual Model

To comprehend the high degree of heterogeneity and hydrogeological complexity within the fracture aquifer system, various techniques, including geophysical and geochemical surveys, along with borehole physical surveys, are employed to gather hydrogeological data, which serve as the foundation for constructing a hydrogeological conceptual model.

A conceptual model provides a description of the geological structure of the aquifer, encompassing features such as stratigraphic structure, rock mechanics, and hydraulic and transport conditions of the aquifer. The hydrogeological conceptual model is essentially a mathematical representation of the subsurface geological structure of the aquifer [26]. The objective behind constructing a conceptual model is to simplify the complexities of the problem and organize the available data. This involves the idea of mapping real-world phenomena into straightforward mathematical models, facilitating subsequent simulations of groundwater flow and solute transport.

Due to the extensive region under consideration, the ECPM or CPM model proves to be more suitable for groundwater flow analysis. In situations where groundwater flow within fractures is significant for a specific region, a hybrid model can be employed. This model not only captures the detail groundwater flow within fractures in a specific region with the DFN model but also effectively considers the overall groundwater flow pattern outside the specific region with the ECPM or CPM models. The hybrid model primarily manifests in two forms: the Layered DFN/ECPM model and the Nested DFN/ECPM model [27]. In various geological environments, sedimentary or crystalline rock layers may coexist in formations containing both porous medium (represented by continuum elements) and fractured medium (treating each fracture as a discrete element). The Layered DFN/ECPM model accommodates this formation property by incorporating both EPM volume elements and DFN (pipe or plate) elements. The Nested DFN/ECPM model utilizes DFN elements at locations where fracture geometry is of utmost interest, such as intersections with boreholes and tunnels. Conversely, they employ ECPM elements based

on equivalent hydraulic parameters derived from Oda upscaling analysis at less sensitive locations. By concurrently utilizing ECPM elements at a large scale and DFN elements at specific borehole locations of immediate interest, the model adequately preserves the essential properties of the DFN hydrogeological model.

*2.3. Groundwater Flow Simulation*

The MAFIC module in FracMan software is a finite element flow model designed to simulate steady-state flow and solute transport in fractured rock. The flow equation in a fracture incorporates the concept of mass conservation and Darcy's law [28]:

$$S_f \frac{\partial h}{\partial t} - T_f \overline{\nabla^2} h = q \tag{3}$$

where $S_f$ is the fracture storativity (-), h is the hydraulic head (L), t is time (T), $T_f$ is the fracture transmissivity ($L^2/T$), q is the source or sink term (L/T), and $\overline{\nabla^2}$ is the two-dimensional Laplace operator.

Similarly, the diffusivity equation for three-dimensional flow in porous media can be written as follows:

$$S_s \frac{\partial h}{\partial t} - K\overline{\nabla^2} h = q \tag{4}$$

where $S_s$ is the specific storage (1/L) and K is the hydraulic conductivity (L/T).

The MAFIC module employs a Galerkin finite element solution scheme to approximate the solution for Equation (3). The finite element approximation to the diffusivity equation in two dimensions is given by the following:

$$\sum_{m=1}^{N} \left[ \int_R \left( T_{f,nm} \overline{\Delta} \xi_n \cdot \overline{\Delta} \xi_m dR \right) h_m \right] + \sum_{m=1}^{N} \left[ \int_R \left( S_{nm} \xi_n \xi_m dR \right) \frac{dh_m}{dt} \right] = \int_R q \xi_n dR$$
$$n = 1, 2, \dots N \tag{5}$$

where $T_f$ is transmissivity ($L^2/T$), S is storativity (-), q is the source flux (L/T), ξ is a linear or quadratic basis function, R is the element area ($L^2$), h is the nodal hydraulic head (L), t is time (T), and N is the number of nodes.

This approximation is also used for modeling flow in the rock matrix. Equation (5) can be expressed in matrix notation as follows:

$$\sum_{m=1}^{N} \left[ A_{nm} h_m + D_{nm} \frac{dh_m}{dt} \right] = Q_n \quad n = 1, 2, \dots N \tag{6}$$

where

$$A_{nm} = \int_R T_{f,nm} \overline{\Delta} \xi_n \cdot \overline{\Delta} \xi_m dR \tag{7}$$

$$D_{nm} = \int_R S_{nm} \xi_n \xi_m dR \tag{8}$$

$$Q_n = \int_R q \xi_n dR \tag{9}$$

The MAFIC module utilizes a backwards difference scheme for which Equation (6) is written as follows:

$$\left[ A + \frac{D}{\Delta t} \right] h^{k+1} = \left[ Q^{k+1} + \frac{D}{\Delta t} h^k \right] \tag{10}$$

where k is the time step. The solution of Equation (10) provides the head values at the end of the time step k + 1.

*2.4. Particle Tracking Algorithm*

The particle tracking algorithm is utilized to represent the concentration of a solute in a solvent, typically groundwater. It does so by defining a finite number of particles, each

having an equal mass and representing a fraction of the total mass of solute in the system. Particles are released at a designated solute source within the simulated region, considering mechanisms such as advection, dispersion, matrix diffusion, and retardation and sorption of various mineral compositions. In this study, the particle tracking algorithm is employed to simulate solute transport, considering advection, diffusion, and dispersion. At each time step, particles move based on the determined advection component and random dispersion component. At the end of each time step, the number of particles reaching sinks is counted and converted to solute concentration.

Particle tracking finds applications in simulating solute transport in both two-dimensional and three-dimensional DFN models [29–31]. The Lagrangian algorithm, commonly used for particle tracking, continuously tracks the physical mass and positions of particles in the flow field, calculating particle counts at any moving position [32]. The origin of particle tracking is early, and its development is mature, making it widely used in hydrogeological analysis software. Examples include the MODPATH module of MODFLOW software (version 2005) [33], the MAFIC module of FracMan software, and the PARTRACK module of DarcyTools software (version 3.4) [34]. Post-particle tracking analysis provides information such as travel time, travel length, and the transport velocity of particles at each time step.

MAFIC employs stochastic particle tracking to simulate solute transport, considering advection, dispersion, and matrix diffusion. The algorithm calculates travel distance based on the velocity field obtained from nodal hydraulic heads at the end of the current time step. For transient flow simulations, where the velocity field changes at each time step, a sufficiently small time step is crucial. For steady-state flow simulations, the change in the velocity field at the time step is ignored, requiring only a single calculation of nodal heads.

Particle motion in triangular elements of the fracture is two-dimensional. For advective flow, velocities in triangular elements are determined by the head field. In the local coordinate system, the triangular element velocities are expressed as follows:

$$V_x = K\frac{h_i - h_j}{X_j} \tag{11}$$

$$V_y = K\frac{X_j(h_i - h_k) + X_k(h_j - h_i)}{Y_k X_j} \tag{12}$$

where K is the hydraulic conductivity of element (L/T) and $h_i$ is the hydraulic head at node i (L).

For dispersion, the following algorithm is applied after calculating the advective pathway. An effective velocity factor, $f_v$, is calculated for the longitudinal direction, i.e., along the flow convective pathway, expressed as follows:

$$f_v = \frac{X_c + X_d}{X_c} \tag{13}$$

where $X_c$ is the convective travel distance (L) and $X_d$ is the longitudinal dispersive travel distance (L).

The deviation angle of transverse dispersion is calculated as follows:

$$\tan\theta = \frac{Y_d}{X_c + X_d} \tag{14}$$

where $Y_d$ is the transverse dispersive travel distance (L).

While incorporating dispersion using Equations (13) and (14), the effective velocity for the longitudinal ($V_l$) and transversal ($V_t$) directions are given by the following:

$$V_l = f_v\sqrt{V_x{}^2 + V_y{}^2} = f_v \cdot V \tag{15}$$

$$V_t = \tan\theta \cdot V_l \tag{16}$$

The effective velocity vectors ($V_l$ and $V_t$) can be decomposed into x- and y-directions. The revised velocities ($V_{x,d}$ and $V_{y,d}$) on the triangular element are then determined as follows:

$$V_{x,d} = V_l \frac{V_x}{V} - V_t \frac{V_y}{V} \tag{17}$$

$$V_{y,d} = V_t \frac{V_x}{V} + V_l \frac{V_y}{V} \tag{18}$$

where $V_{x,d}$ is the particle velocity in the x-direction with dispersion correction, $V_{y,d}$ is the particle velocity in the y-direction with dispersion correction, $V_x$ is the particle velocity in the x-direction without dispersion, $V_y$ is the particle velocity in the y-direction without dispersion, and V is the total velocity without dispersion, expressed by Equation (19):

$$V = \sqrt{V_x{}^2 + V_y{}^2} \tag{19}$$

Introducing Equation (15) to (18), the revised velocity becomes

$$V_{x,d} = f_v\left(V_x - \tan\theta V_y\right) \tag{20}$$

$$V_{y,d} = f_v\left(\tan\theta V_x + V_y\right) \tag{21}$$

Relative to the starting coordinates ($X_0$ and $Y_0$), the new particle coordinates after $\Delta t$ time are calculated:

$$X = X_0 + V_{x,d} \cdot \Delta t \tag{22}$$

$$Y = Y_0 + V_{y,d} \cdot \Delta t \tag{23}$$

where $X_0$ and $Y_0$ are the particle's initial coordinate at the current element.

The groundwater flow field and velocity field can be calculated within the finite element grid, and then the particles can be tracked in the velocity field to obtain their travel path. FEM combined with particle tracking can be widely used for flow visualization or solute transport analogies, and a combination of travel paths marked at regular time intervals provides visual information about travel time and flow direction.

In this study, the MAFIC module uses FEM to solve groundwater flow and particle tracking. The result of the flow field is the pressure value at the node of the element, and the velocity at the element is calculated by interpolation. However, the spatial interpolation of velocity may introduce errors into particle tracking, which possibly cause different velocities, travel paths, and travel times of particles, or even particles getting stuck in the middle and being unable to travel anymore, especially for relatively coarse grids [35]. Despite this disadvantage, FEM is rather popular due to its capability of handling complex geometry.

## 3. In Situ Tracer Experiment and Site Descriptive Model

### 3.1. In Situ Tracer Experiment

The experimental site is situated in the slope terrain of Taoyuan City, Taiwan, characterized by a composition of gravel formation and Cholan formation, as illustrated in Figure 1a. In Figure 1a, the yellow region represents the gravel formation exposed on the ground surface, and the brown region represents the Cholan formation exposed on the ground surface. The geological structure predominantly consists of the Cholan formation, dating from the Pliocene to the Pleistocene, and the Quaternary Pleistocene gravel formation. The Cholan formation, named by the Japanese geologist Torii in 1935, is widespread in western Taiwan. The average thickness of the Cholan formation is 2000 m. It primarily comprises sandstone, siltstone, shale, and mudstone. The geological profile along the C-C' line in the experimental site shows that the rock plate elevation in this region is approximately

195 m (Figure 1b). Above it are the gravel layer and overburden layer, while below it is the Cholan layer consisting of argillaceous sandstone, as depicted in Figure 1b.

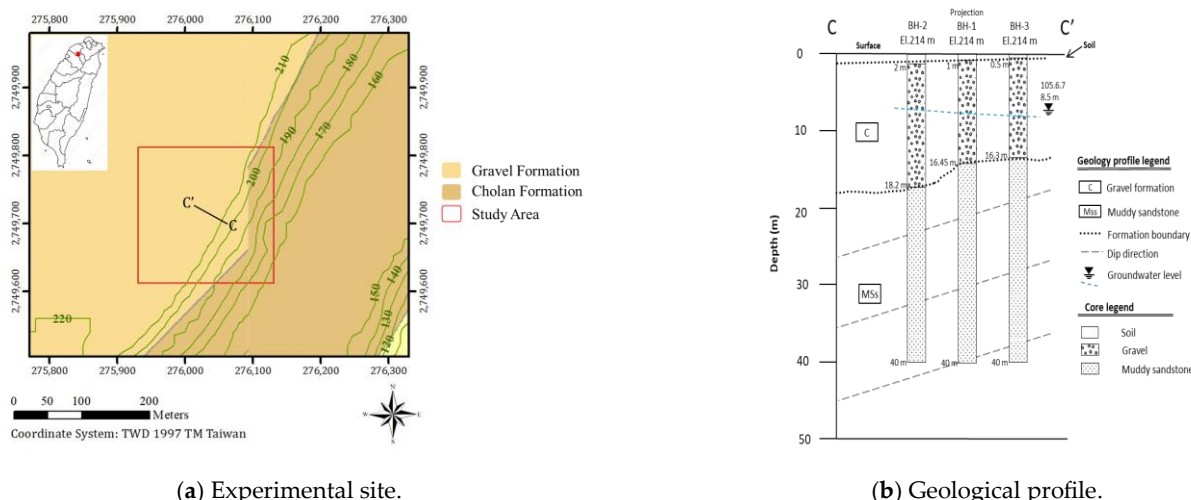

(**a**) Experimental site.

(**b**) Geological profile.

**Figure 1.** Description and geology of experimental site: (**a**) location and scope of experimental site, the red point on the map represents the location of experimental site; (**b**) geology of C-C' profile.

This study conducted an in situ tracer test in this experimental site to gain insights into the actual dynamics of solute transport. In the tracer experiment, sodium bromide served as the tracer, with 40 kg of sodium bromide solute mixed into 160 L of aqueous solution to create a sodium bromide solution. Additionally, 320 L of water was injected as a supplementary agent to ensure the tracer could thoroughly flow out of the well and enter the aquifer. The initial tracer concentration can be calculated as follows:

$$\text{Initial tracer concentration} = \frac{40 \text{ Kg (sodium bromide)}}{480 \text{ L (water)}} = 8.333 \times 10^4 \text{ ppm}$$
$$= 8.333 \times 10^7 \text{ ppb} \tag{24}$$

The injection location is an upstream release well, with the observation located at a natural outflow on the slope. Tracer release commenced at 10:00 a.m. and was introduced into the groundwater as a step function, as shown in Figure 2a. The material composition of the formation medium significantly influences the tracer transport behavior. Tracers are transported through formation media via mechanisms such as diffusion, advection, and dispersion. In the early stage of tracer reception, the concentration gradually increases due to advection and dispersion mechanisms. Subsequently, tracers are influenced by the diffusion mechanism, causing the concentration to decrease continuously. Water samples were collected from an observation point on the downstream slope for manual measurement of the tracer concentration. As the concentration peaked between 2:00 p.m. and 3:00 p.m., manual measurements concluded at 5:00 p.m. The results of measuring the tracer concentration at the observation location are shown in Figure 2b. The findings reveal that tracer concentration can be detected approximately 2.833 h after release, reaching a maximum tracer concentration of about 3060.343 ppb reached at approximately 4.5 h.

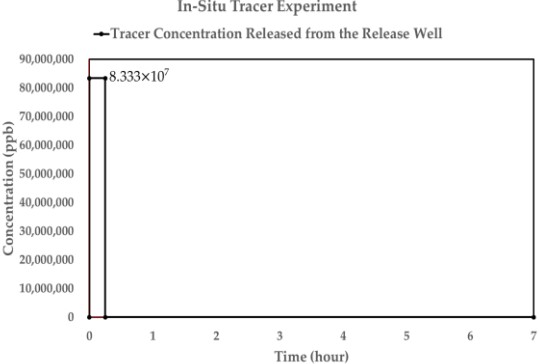

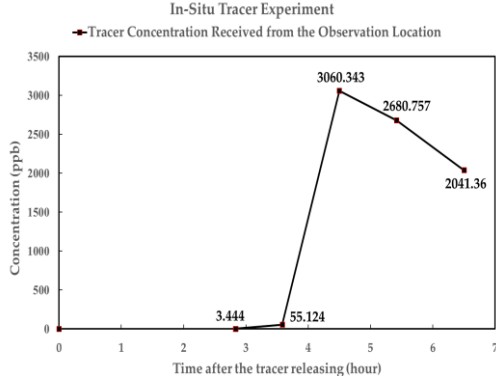

| (**a**) Concentration of released tracer. | (**b**) Concentration of received tracer. |

**Figure 2.** Tracer concentrations for in situ tracer experiment. Values in figure are measured values from in situ tracer experiment.

### 3.2. Site Descriptive Model

Based on the borehole drilling survey results, this study determined that the argillaceous sandstone of the Cholan formation is the bedrock, with its surface at an elevation of approximately 195 m. Above the bedrock, there is a gravel formation above it, extending from about 220 m to 195 m. The argillaceous sandstone of the Cholan formation lies below the surface elevation of 195 m. To supplement the survey data, five tests were conducted in this study, including the bedrock leakage test (Legeon test), the borehole packer test, borehole photography, the borehole deformation test, and P-S logging. The results show that the average hydraulic conductivity of the gravel layer is approximately $2.3 \times 10^{-4}$ m/s, and the average porosity is $5 \times 10^{-3}$ (-). For the Cholan formation, the average hydraulic conductivity is $9.70 \times 10^{-9}$ m/s, and the average porosity is $3.78 \times 10^{-3}$ (-). The hydraulic conductivity of the formation is shown in Table 1. Since the Cholan formation is an argillaceous sandstone, it is commonly treated as a porous medium in simulation work. However, a total of 42 fractures were collected from the borehole photography, combined with the drill core records and geological statistics; the fracture sets in the Cholan formation were estimated to be Joint-1 (J1), Bedding (B), and Joint-2 (J2), and their DFN parameters are shown in Table 2. Groundwater level measurement results from over several years show that the average groundwater level is 7 m above the bedrock, and the hydraulic gradient is estimated to be 0.01.

In this study, FracMan software (version 7.9) was used to develop geological hybrid models and perform the associated simulations. The numerical simulation involved the use of particle tracking within the MAFIC module to correlate with the results from the in situ tracer experiment. The simulation range, encompassing the release well and the natural outflow of the slope, was defined as a 100 m span in the x-direction, a 100 m span in the y-direction, and an elevation range from 216 m to 158 m along the z-direction. The elevation of the bedrock was set at 195 m, with the gravel formation above the Cholan formation of argillaceous sandstone. The conceptual model of the experimental site is shown in Figure 3. Examination of borehole core images reveals that, while the region between elevations of 195 m and 190 m is bedrock, it exhibits a fragmented nature and a texture resembling a porous medium.

**Table 1.** Hydraulic conductivity of gravel formation and Cholan formation.

| Rock Matrix | Hydraulic Conductivity (m/s) | | |
| --- | --- | --- | --- |
| | Minimum | Average | Maximum |
| Gravel formation | $1.15 \times 10^{-6}$ | $2.3 \times 10^{-4}$ | $1.70 \times 10^{-3}$ |
| Cholan formation | $5.65 \times 10^{-11}$ | $9.7 \times 10^{-9}$ | $3.08 \times 10^{-8}$ |

**Table 2.** DFN parameters of the Cholan formation.

| DFN Parameter | Parameter Value |
|---|---|
| Fracture cluster (Trend/Plung/Kappa/$P_{32,rel}$) | Set-1: J1 = 147/62/Fish distribution ($\kappa$ = 14.43)/$P_{32,rel}$ = 55.0% |
| | Set-2: B = 331/13/Fish distribution ($\kappa$ = 50.14)/$P_{32,rel}$ = 32.5% |
| | Set-3: J2 = 33/39/Fish distribution ($\kappa$ = 28.49)/$P_{32,rel}$ = 12.5% |
| Fracture Intensity, $P_{32}$ [1] | $P_{32}$ ($r^2 > r_0$) $\approx P_{10,corr}$ = 1.71 m$^{-1}$ |
| Fracture Size | Power law: $k_r$ = 2.83 [3], $r_0$ = 0.1 m [4], $r_{max}$ = 100 m [5] |
| Fracture Location | Stationary random (Poisson) process |
| Transmissivity | T = 1.0 $\times 10^{-9}$ $r^{0.7}$ (m$^2$/s) |
| Aperture | Doe law: e = 0.5$T^{0.5}$ (m) |

Notes: [1] $P_{32}$: total area of fractures per unit volume of rock mass (volumetric intensity, m$^{-1}$). [2] r: the fracture radius. [3] $k_r$: the exponent of fractal dimension, or the so-called fracture radius scaling exponent. [4] $r_0$: the minimum radius value. [5] $r_{max}$: the maximum radius value.

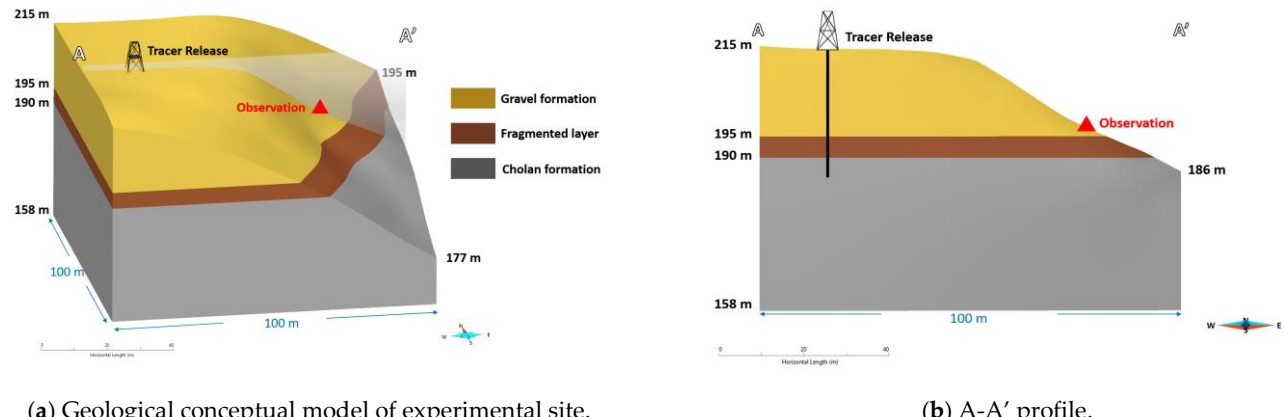

(**a**) Geological conceptual model of experimental site.　　　　　　　　　　(**b**) A-A′ profile.

**Figure 3.** Conceptual model of the experimental site.

The CPM model can be applied to the gravel formation with an elevation above 195 m according to its geological characteristics. Based on the investigation, there are DFN parameters in the Cholan formation below the elevation of 195 m. The geological models that can be discussed in the Cholan formation are DFN, CPM, ECPM, and Hybrid DFN/ECPM models. Four associated hydrogeological cases were constructed by FracMan software in this study: Case-1: CPM-CPM model; Case-2: CPM-DFN model; Case-3: CPM-ECPM model; and Case-4: CPM-ECPM-DFN model, as shown in Figure 4. The legend shows the hydraulic conductivity of the matrix for CPM and ECPM from $1 \times 10^{-12}$ m/s to $2.3 \times 10^{-4}$ m/s. Case-3 and Case-4 both have the equivalenthydraulic conductivity and equivalent porosity as the ECPM model obtained by Oda upscaling analysis from the DFN model. In Case-3, the hydraulic conductivity of the ECPM model ranged from $1.001 \times 10^{-12}$ m/s to $5.055 \times 10^{-8}$ m/s, with an average of $7.345 \times 10^{-9}$ m/s, and the porosity ranged from $3.139 \times 10^{-11}$ to $2.152 \times 10^{-2}$ (-), with an average of $3.32 \times 10^{-3}$ (-). In Case-4, the hydraulic conductivity ranged from $1.045 \times 10^{-12}$ m/s to $3.273 \times 10^{-8}$ m/s, the average was $6.425 \times 10^{-9}$ m/s, and the porosity ranged from $4.867 \times 10^{-11}$ to $4.999 \times 10^{-3}$ (-), and the average was $2.471 \times 10^{-3}$ (-). The hydraulic conductivity and porosity results of the Oda upscaling analysis are not far from the in situ measurements of the Cholan argillaceous sandstone. The grids of the CPM or ECPM models we established in this study were all hexahedral, and the grid volume ranged from 0.2 to 1.8 m$^3$, while the grid on the fracture plane was triangular, and the grid area ranged from 0.1 to 1.535 m$^2$.

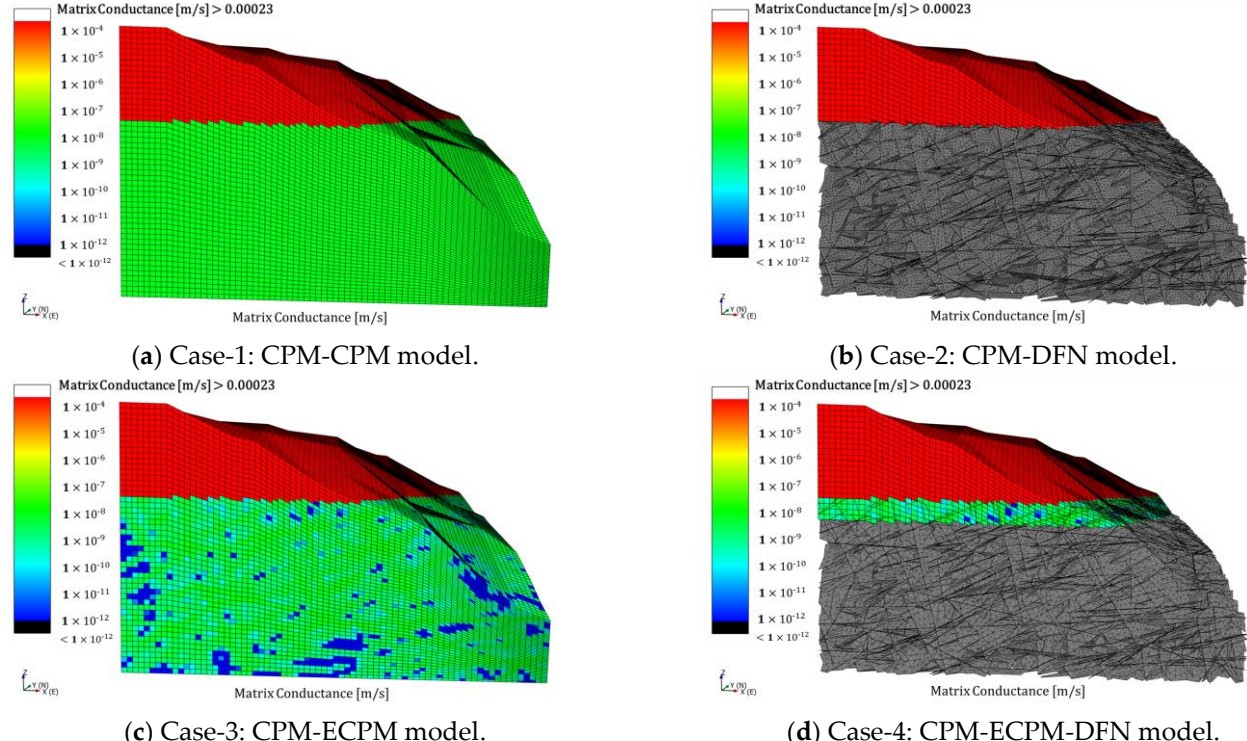

(a) Case-1: CPM-CPM model.

(b) Case-2: CPM-DFN model.

(c) Case-3: CPM-ECPM model.

(d) Case-4: CPM-ECPM-DFN model.

**Figure 4.** Hydrogeological conceptual model for the experimental site. The legend shows the hydraulic conductivity of porous media in the range of $1 \times 10^{-12}$ to $2.3 \times 10^{-4}$ m/s.

The groundwater hydraulic gradient in this region is approximately 0.01 based on the results of the site investigation. A high water table is set on the western boundary of the model, with the groundwater table positioned 7 m above the bedrock surface. The observation point of the slope is the natural outflow location, where the atmospheric pressure is set. Since there is no natural outflow in the region except the observation position, the rest of the lateral boundaries are set as the no-flow boundary condition. The no-flow boundary is also set at the bottom of the model at an elevation of 158 m. Typically, the dispersivity setting is within the range of 1/100 to 1/10 of the transport length. Considering that the shortest distance from the release well to the observation point is 64.8 m, this study assumes longitudinal dispersivity and transverse dispersivity to be 5 m and 2.5 m, respectively. The diffusion coefficient of groundwater is $1.0 \times 10^{-9}$ m$^2$/s.

## 4. Results and Discussion

The simulation results of the steady-state groundwater flow of each case are shown in Figures 5a–8a, respectively. The legend shows that the hydraulic head ranges from 0 to 7 m. A total of 400,000 particles are released in the release well after the steady-state groundwater flow is obtained. Since an $8.333 \times 10^7$ ppb concentration of tracer is released in the in situ tracer experiment, each particle represents a concentration of 208.325 ppb when the solute transport is simulated by the particle tracking in the numerical simulation. The MAFIC module performs the particle tracking simulation, and the results of particle end location, travel time, travel length, and transport velocity at each time step are obtained. This information enables the conversion of the relationship between the number of particles arriving and travel time to the relationship between concentration and time. This correlation is then utilized to discuss the results between the numerical simulation and the in situ tracer experiment.

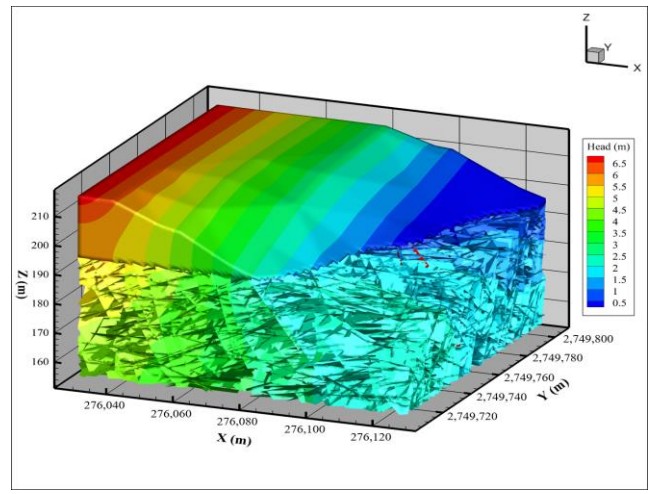

(**a**) Steady-state groundwater flow (head range is 0 to 7 m).

(**b**) Travel length of particles (length range is 64.8 to 74.05 m).

(**c**) Travel time of particles (time range is 3.06 to 13.64 h).

(**d**) Number of particles and travel time.

**Figure 5.** Steady-state groundwater flow and particle tracemap of Case-1.

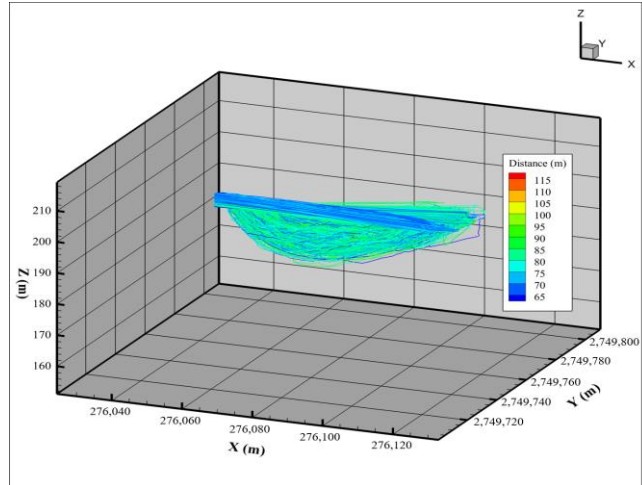

(**a**) Steady-state groundwater flow (head range is 0 to 7 m).

(**b**) Travel length of particles (length range is 64.8 to 113.79 m).

**Figure 6.** *Cont*.

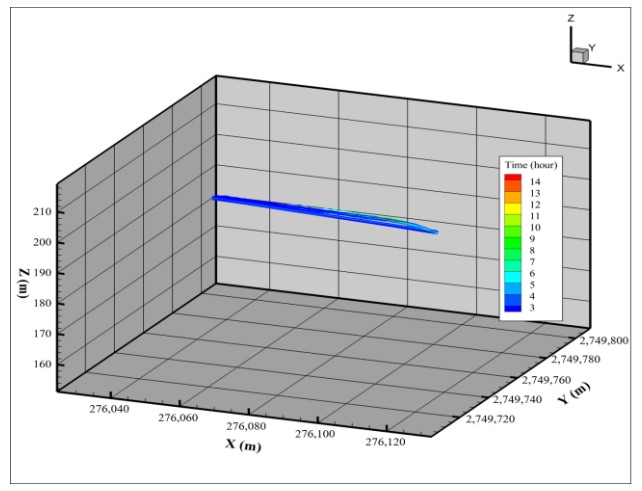

(**c**) Travel time of particles (time range is 2.92 to 14.36 h).

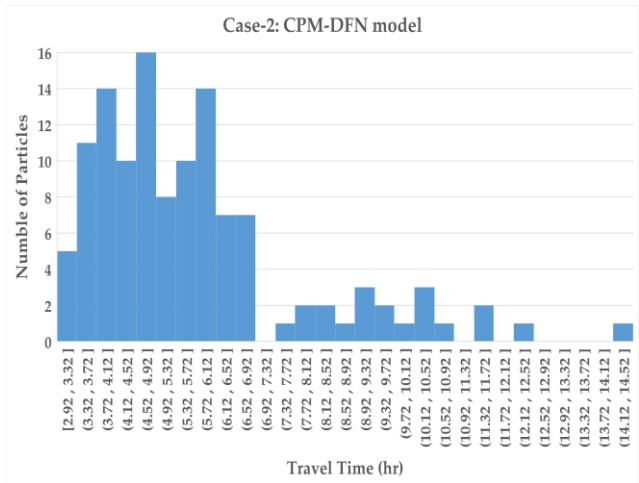

(**d**) Number of particles and travel time.

**Figure 6.** Steady-state groundwater flow and particle tracemap of Case-2.

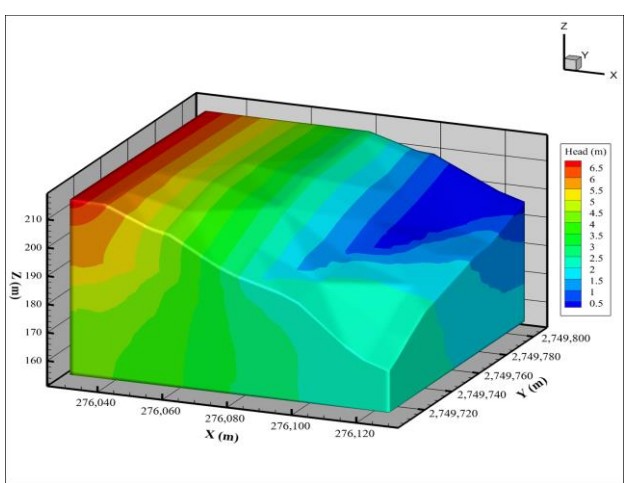

(**a**) Steady-state groundwater flow (head range is 0 to 7 m).

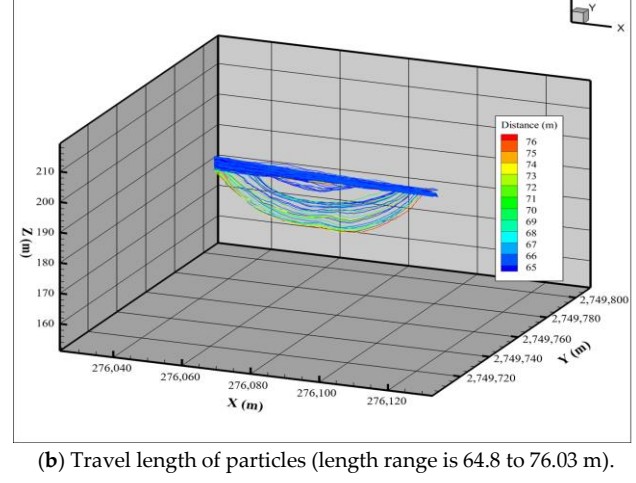

(**b**) Travel length of particles (length range is 64.8 to 76.03 m).

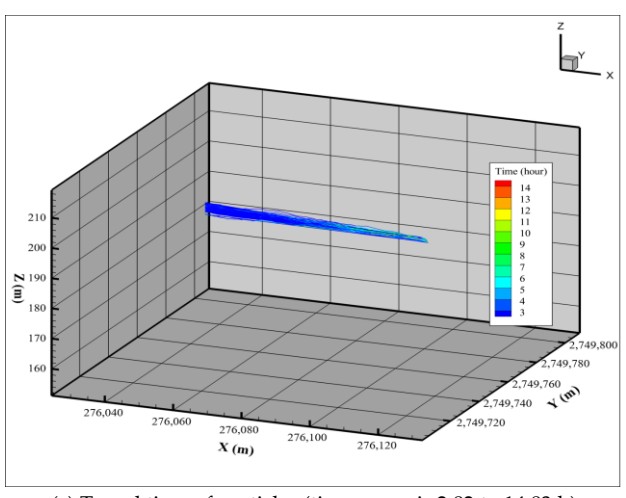

(**c**) Travel time of particles (time range is 2.82 to 14.83 h).

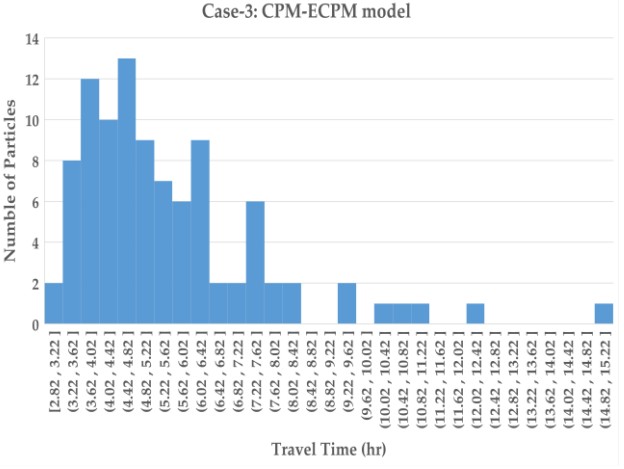

(**d**) Number of particles and travel time.

**Figure 7.** Steady-state groundwater flow and particle tracemap of Case-3.

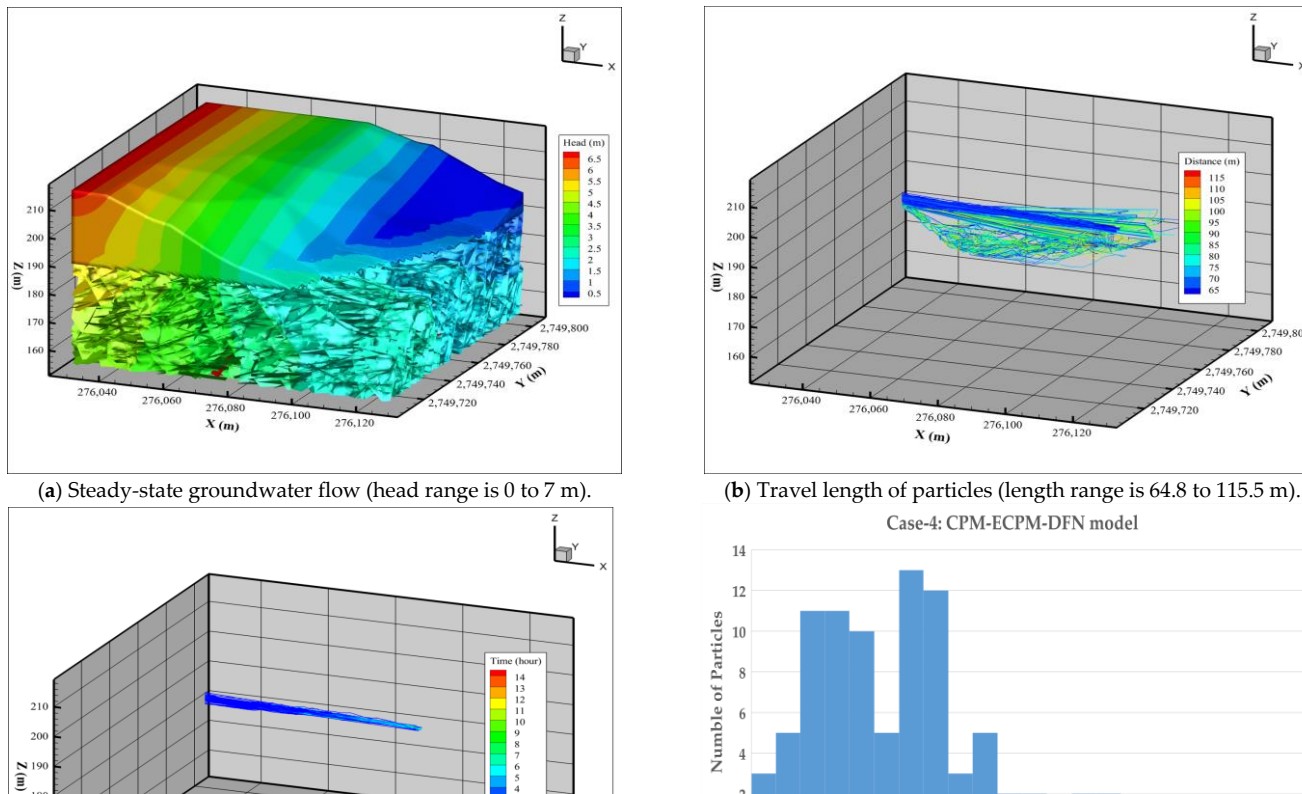

(**a**) Steady-state groundwater flow (head range is 0 to 7 m).

(**b**) Travel length of particles (length range is 64.8 to 115.5 m).

(**c**) Travel time of particles (time range is 2.92 to 12.07 h).

(**d**) Number of particles and travel time.

**Figure 8.** Steady-state groundwater flow and particle tracemap of Case-4.

The following sections will discuss the simulation results of various hydrogeological conceptual models, including particle pathways, travel lengths, and travel times, and their comparison with the results of the tracer test.

### 4.1. Particle Travel Length of Different Models

Since not every particle smoothly reaches the natural outflow position, some particles are stuck due to unconnected pathways, called dead ends. Particles that do not reach the outflow point are excluded from subsequent track information calculations. Given that the shortest distance from the release well to the natural outflow position is approximately 64.8 m, the initial screening involves considering only those particles with a minimum travel length greater than 64.8 m. The outcomes of the first screening are shown in Figures 5b–8b. Under this first screening criterion (as shown in Table 3), in Case-1, 938 particles met the criterion with a maximum travel length of 74.05 m; in Case-2, 556 particles met the criterion with a maximum travel length of 113.79 m; in Case-3, 228 particles met the criterion with a maximum travel length of 76.03 m; in Case-4, 472 particles met the criterion with a maximum travel length of 115.50 m.

Although Case-1 and Case-3 are both continuous porous medium models, the significant difference between them lies in the variance of hydraulic conductivity and porosity. In Case-1, hydraulic conductivity and porosity values for the Cholan formation are both single values, resulting in a homogeneous distribution of hydraulic parameters. This phenomenon makes a relatively large number of particles able to smoothly reach the natural outflow position. However, in Case-3, the equivalent parameters are obtained after upscaling

analysis of the DFN model and are not single values. The equivalent parameters, such as hydraulic conductivity and porosity, depend on the intensity of fracture distribution. If there are more fractures in the matrix, the equivalent parameters after upscaling will be larger. Conversely, if there are fewer fractures in the matrix, the equivalent parameters will be smaller. This phenomenon leads to particles migrating through a complex domain and causes significant variation in particle traces. Compared with the results of Case-1, some particles may become trapped in dead ends or need to take detours before moving forward again in Case-3. Therefore, the number of particles that meet the screening criteria is relatively smaller, and the travel length is relatively longer.

**Table 3.** Results of the first screening for particle pathway (the travel length must be greater than 64.8 m).

| Hydrogeological Conceptual Model | Case-1: CPM-CPM Model | Case-2: CPM-DFN Model | Case-3: CPM-ECPM Model | Case-4: CPM-ECPM-DFN Model |
|---|---|---|---|---|
| Number of particles | 938 | 556 | 228 | 472 |
| Minimum length (m) | 64.8 | 64.8 | 64.8 | 64.8 |
| Maximum length (m) | 74.05 | 113.79 | 76.03 | 115.5 |

Both Case-2 and Case-4 are hybrid models incorporating the continuous porous medium model and the DFN model. The connectivity of fracture systems significantly influences particle movement within fractures. If the fractures are poorly connected, particles may become easily trapped or interrupted. Conversely, if the fracture systems exhibit good connectivity, particles will have a greater opportunity to traverse multiple fractures, resulting in longer travel distances and times. In comparison to Case-1, the influence of fracture connectivity within the Cholan formation leads to a relatively smaller number of particles meeting screening criteria and longer travel lengths in both Case-2 and Case-4.

*4.2. Particle Travel Time of Different Models*

Considering the maximum measurement time of the in situ tracer experiment, the second screening criterion is that the maximum travel time should be less than 15 h. This study conducted the second screening based on the results that passed the first screening, and the outcomes of the second screening are shown in Figures 5c–8c. Under this second screening criterion (as shown in Table 4), in Case-1, 101 particles met the criterion with travel times ranging from 3.06 to 13.64 h; in Case-2, 122 particles met the criterion with travel times ranging from 2.92 to 14.36 h; in Case-3, 97 particles met the criterion with travel times ranging from 2.82 to 14.83 h; in Case-4, 91 particles met the criterion with travel times ranging from 2.90 to 12.07 h.

**Table 4.** Results of the second screening for particle pathway (the travel time must be less than 15 h).

| Hydrogeological Conceptual Model | Case-1: CPM-CPM Model | Case-2: CPM-DFN Model | Case-3: CPM-ECPM Model | Case-4: CPM-ECPM-DFN Model |
|---|---|---|---|---|
| Number of particles | 101 | 122 | 97 | 91 |
| Minimum time (hours) | 3.06 | 2.92 | 2.82 | 2.90 |
| Maximum time (hours) | 13.64 | 14.36 | 14.83 | 12.07 |

The results show that particle tracking occurs predominately in the gravel layer along the interface between the gravel formation and the Cholan formation. The concentration received within 15 h reflects the migration in the gravel formation during the in situ tracer experiment. If tracers migrate through the Cholan formation, it may take an extended period to observe the concentration response.

### 4.3. Transport Resistance of Different Models

The transport resistance, often referred to as the "F-factor", is a parameter that describes the potential for retention and retardation of particle transport along a travel path [36–38]. This parameter is influenced by formation properties and flow-related characteristics, and it can be estimated based on the velocity of particle travel within the formation. Transport resistance is typically inversely to flow velocity, meaning that higher resistance corresponds to lower travel velocity.

This study calculates the transport resistance from the travel time and travel length of each particle's pathway. Figure 9a presents the boxplot of transport resistance for different models, derived from the results of the first screening, which was based on particle travel length. Case-1 and Case-3 both represent porous medium models, where particle movement is primarily driven by the difference in groundwater pressure. Case-2 and Case-4 involve fracture models, where the complexity of fracture connections and the varying properties of the fractures themselves (such as aperture and transmissivity) influence the travel length and time of pathways.

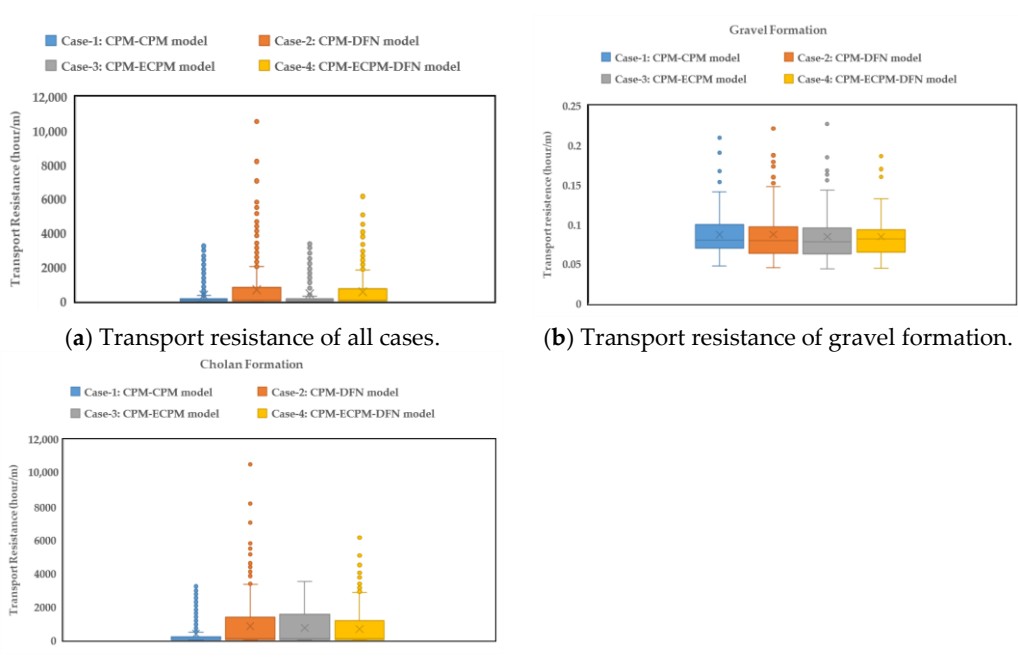

(**a**) Transport resistance of all cases.

(**b**) Transport resistance of gravel formation.

(**c**) Transport resistance of Cholan formation.

**Figure 9.** Transport resistance of each model and each formation.

From the results of the second screening based on the particle travel time, it is evident that all particles exclusively traverse the gravel formation. The boxplot of transport resistance for different models is shown in Figure 9b. In the gravel formation, both the median and mean of transport resistance are approximately 0.079 h/m and 0.086 h/m, respectively. This distribution remains consistent across all models, indicating uniform transport resistance within the gravel layer in this experimental site.

From the pathway of particle travel in the Cholan formation for each model, the boxplot of transport resistance for different models is shown in Figure 9c. Given the heterogeneous nature of the Cholan formation itself, the DFN model or the ECPM model with upscaling by the DFN model can better preserve this heterogeneity compared to the CPM model. Disregarding the results of Case-1, the average transport resistance values within the Cholan formation from Case-2, Case-3, and Case-4 are nearly identical, averaging approximately 748.71 h/m.

### 4.4. Comparative Analysis of Numerical Simulation and Tracer Test Results

Figures 5d–8d show the number of particles arriving and their travel time for each case. Given that each particle represents a tracer concentration of 208.325 ppb, the results have been converted into concentration–time curves and compared with the in situ tracer breakthrough curve, shown in Figure 10. Table 5 shows the error comparison results between the numerical simulation and tracer experiment for the time of the earliest received concentration, the maximum received concentration, and the time of the maximum received concentration. The results indicate that the simulation result of Case-3 at the time of the earliest received concentration is the most accurate, while the simulation result of Case-1 at the maximum received concentration performs optimally. The simulation results of Case-2 and Case-3 at the time of the maximum received concentration are the most accurate. These findings show that the combination of the CPM-ECPM geological conceptual model of Case-3 provides a reasonable representation of the local geological structure. The hydraulic properties of the Cholan formation, influenced by the distribution of fractures, exhibit heterogeneity. The ECPM model is more suitable for illustrating the heterogeneity of the Cholan formation.

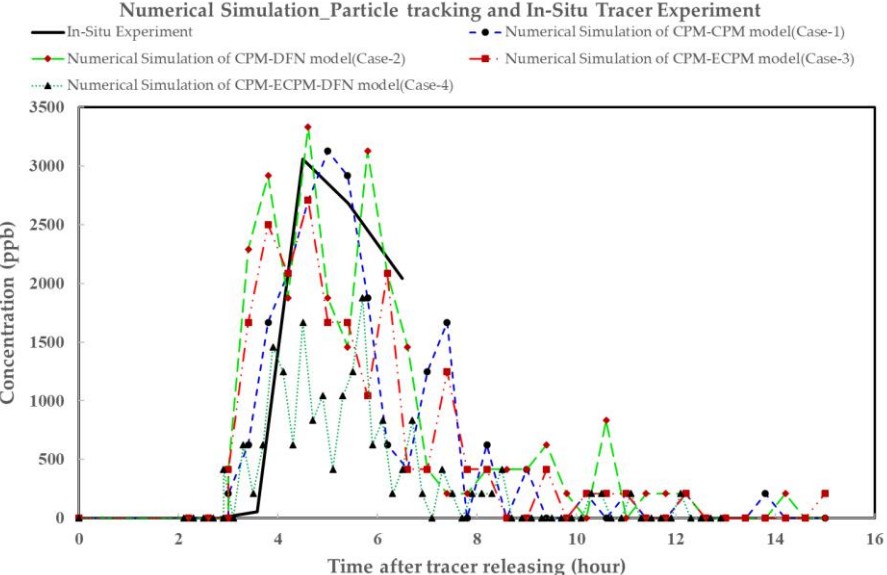

**Figure 10.** Results of numerical simulation in each case and in situ tracer experiment.

**Table 5.** Comparison of the numerical simulation of each case and the in situ tracer experiment.

| Hydrogeological Conceptual Model | In Situ Tracer Experiment | Numerical Simulation: Particle Tracking Algorithm | | | |
|---|---|---|---|---|---|
| | | Case-1 CPM-CPM Model | Case-2: CPM-DFN Model | Case-3: CPM-ECPM Model | Case-4: CPM-ECPM-DFN Model |
| The earliest time to receive the concentration (hours) | 2.833 | 3.06 | 2.92 | 2.82 | 2.90 |
| Differences from tracer test result (hours) | - | +0.227 | +0.087 | −0.013 | +0.067 |
| Maximum received concentration (ppb) | 3060.343 | 3124.875 | 3333.200 | 2708.225 | 1874.925 |
| Differences from tracer test result (ppb) | - | +64.532 | +272.857 | −352.118 | −1185.418 |
| Time to receive maximum concentration (hours) | 4.50 | 5.00 | 4.60 | 4.60 | 5.70 |
| Differences from tracer test result (hours) | - | +0.5 | +0.1 | +0.1 | +1.2 |

*4.5. Sensitivity Analysis of Rock Hydraulic Conductivity in Various Formation*

As mentioned in Section 4.5, this study found that the CPM-ECPM hybrid model provides a reasonable depiction of the geological structure at the experimental site through a comparison of numerical simulation and tracer test results. Given that the hydraulic properties of the Cholan formation are influenced by fracture distribution, the ECPM model emerged as more suitable for illustrating the heterogeneity of the Cholan formation. Within the ECPM grid, hydraulic conductivity is derived from the DFN model via upscaling analysis. In instances where no fractures are present in the ECPM grid, hydraulic conductivity is determined by the rock mass itself.

In this experimental site, the hydraulic conductivity of the gravel and Cholan formation exhibits maximum, minimum, and average values, as outlined in Table 1. The description of parameter variability in this study is shown in Table 6, which was combined into 9 cases to calculate particle tracking and concentration. This study focuses on Case-3's CPM-ECPM model and integrates hydraulic conductivity coefficients from both the gravel and Cholan formations in each case. In the discussed scenario, if the ECPM grid within the Cholan formation represents integral rock, hydraulic conductivity values are quoted accordingly. Conversely, if it signifies fractured rock, hydraulic conductivity retains its original value obtained through upscaling analysis.

**Table 6.** Case description of sensitivity analysis.

| Case | Hydraulic Conductivity (m/s) | | |
| --- | --- | --- | --- |
| | Gravel Formation | Cholan Formation | |
| | | Integral Rock [1] | Fractured Rock [2] |
| Case-3 | $2.3 \times 10^{-4}$ | $K_{integral\ rock} = K_{min}{}^{3} = 5.65 \times 10^{-11}$ | |
| Case-3_1 | $1.15 \times 10^{-6}$ | $K_{integral\ rock} = K_{min} = 5.65 \times 10^{-11}$ | |
| Case-3_2 | $1.70 \times 10^{-3}$ | $K_{integral\ rock} = K_{min} = 5.65 \times 10^{-11}$ | |
| Case-3_3 | $2.3 \times 10^{-4}$ | $K_{integral\ rock} = K_{avg}{}^{3} = 9.7 \times 10^{-9}$ | |
| Case-3_4 | $2.3 \times 10^{-4}$ | $K_{integral\ rock} = K_{max}{}^{3} = 3.08 \times 10^{-8}$ | |
| Case-3_5 | $1.15 \times 10^{-6}$ | $K_{integral\ rock}\ K_{avg} = 9.7 \times 10^{-9}$ | |
| Case-3_6 | $1.15 \times 10^{-6}$ | $K_{integral\ rock} = K_{max} = 3.08 \times 10^{-8}$ | |
| Case-3_7 | $1.70 \times 10^{-3}$ | $K_{integral\ rock} = K_{avg} = 9.7 \times 10^{-9}$ | |
| Case-3_8 | $1.70 \times 10^{-3}$ | $K_{integral\ rock} = K_{max} = 3.08 \times 10^{-8}$ | |

Notes: [1] The grid in the ECPM model has no fractures, and this grid is regarded as integral rock. [2] The grid in the ECPM model has fractures, and this grid is regarded as fractured rock. [3] $K_{min}$, $K_{avg}$ and $K_{max}$ are the hydraulic conductivity of the Cholan formation measured through experiments.

This study considers the maximum concentration value and the corresponding time of occurrence for sensitivity analysis, and compares them with the tracer results (Figure 11). Instances where the hydraulic conductivity of the gravel formation adopts the minimum value result in maximum concentrations occurring later (well beyond 15 h) and are therefore not depicted in Figure 11. Conversely, when the hydraulic conductivity of the gravel formation adopts the average value, maximum concentration and time align more closely with the tracer test results. When the hydraulic conductivity of the gravel formation adopts the maximum value, the maximum concentration occurs earlier.

Compared to the Cholan formation, changes in hydraulic conductivity within the gravel formation exhibit significantly greater sensitivity to tracer transport at this site. Thus, employing the average hydraulic conductivity for the gravel formation proves more suitable for aligning with tracer results.

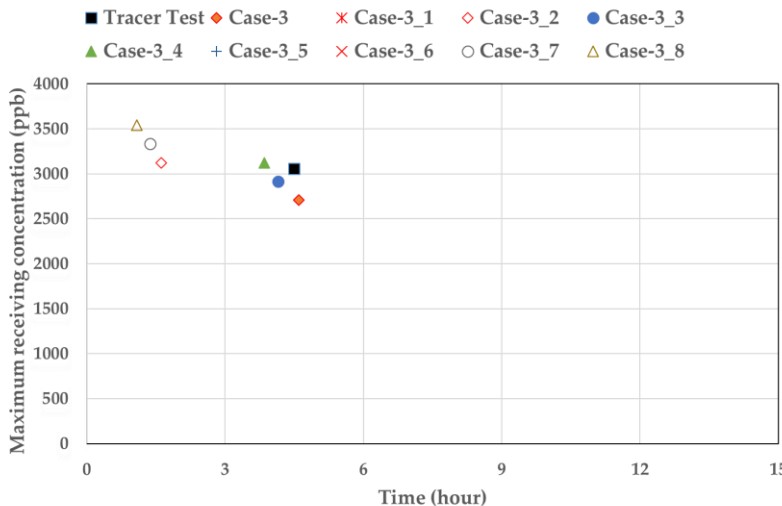

**Figure 11.** Sensitivity of the hydraulic conductivity of the aquifer to tracer concentration. Note: The results of Case-3_1, Case-3_5, and Case-3_6 are not shown in the figure because their maximum concentration values occurred after 15 h.

## 5. Conclusions

In this study, sodium bromide was chosen as the tracer due to its minimal interference with existing elements in the composite formation of gravel and Cholan in northwest Taiwan. It was used in a small-scale region, resulting in a well-defined tracer concentration breakthrough curve. Numerical simulation must account for the uncertainties present in the current situation. It involves constructing an appropriate hydrogeological conceptual model and applying reasonable groundwater flow theory to simulate solute transport. The Cholan formation, characterized by argillaceous sandstone, is commonly treated as a porous medium. However, fractures were identified through borehole photography and combined with drill core records and geological statistics to define the transport characteristics of the fractured medium. To establish a suitable conceptual model for the specialized Cholan formation, four hydrogeological conceptual models, based on the characteristics of the in situ geological formation, were constructed in this study. The numerical simulation results of groundwater flow and the particle tracking algorithm were then compared to the in situ tracer experiment. Tracer concentrations were artificially measured within 7 h during the in situ tracer experiment, with the peak concentration recorded after 4.5 h. In the particle tracking algorithm, particles traced within a travel time of less than 15 h were consistently located in the gravel formation, moving along the interface between the gravel formation and the Cholan formation. If tracers migrated through the Cholan formation, concentration responses could be observed over an extended period.

When the in situ tracer test results are integrated into numerical simulations, it can be confirmed that the combination of the CPM-ECPM geological conceptual model can reasonably describe the local geological structure. The ECPM model proved more suitable for illustrating the heterogeneity of the Cholan formation, as the hydraulic properties of the Cholan formation are influenced by the distribution of fractures. Conversely, when the numerical simulation results are integrated into the in situ tracer test, they indirectly reflect the long-term trend of tracer transport at this site, based on strong evidence derived from considering various models and comparing them with in situ tracer tests.

**Author Contributions:** Conceptualization, C.-Z.T., P.Y. and Y.-C.Y.; methodology, C.-Z.T. and P.Y.; software, C.-Z.T.; validation, Y.-C.Y.; resources, L.-G.C. and H.-H.T.; data curation, P.Y.; writing—original draft preparation, C.-Z.T.; writing—review and editing, Y.-C.Y.; visualization, C.-Z.T. and Y.-C.Y.; supervision, L.-G.C. and H.-H.T.; project administration, L.-G.C. and H.-H.T. All authors have read and agreed to the published version of the manuscript.

**Funding:** This research was funded by the National Science and Technology Council, Taiwan, grant number 112-3111-Y-042A-012.

**Data Availability Statement:** All data are presented in this article in the form of figures and tables.

**Conflicts of Interest:** The authors declare no conflicts of interest.

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
