# Peer review of "Verification of Particle Tracking and In Situ Tracer Experiment for the Gravel and Cholan Formation Composite in Northwest Taiwan"

_water, doi:10.3390/w16081101_

Round 1

Reviewer 1 Report

Comments and Suggestions for Authors

This study verifies the advantages of employing particle tracking to assess tracer transport in a specific groundwater system case study. While the paper presents intriguing findings, the introduction lacks evident novelty. Is the case study itself considered novel? If so, what makes it unique? Have other studies been conducted in the selected region of interest? Do the authors offer new insights into the region? Once these questions (or others) are posed in the introduction, the discussion should provide answers to them.

Minor comments
Remove the bold formatting from equations unless they represent matrices or vectors. If they are standard equations, they should not be bold.
Ensure consistency in the formatting of variables in equations. If S_f is not intended to be bold, make sure it matches the formatting of other variables.
Consider adding a comment regarding the selected modeling framework, specifically discussing the choice of FEM (Finite Element Method) combined with Particle Tracking. Additionally, it may be beneficial to explore alternative methods such as SPH (Smoothed Particle Hydrodynamics). For instance, references such as "An alternative smooth particle hydrodynamics formulation to simulate chemotaxis in porous media" and "A meshless method to simulate solute transport in heterogeneous porous media" could provide insights into the applicability and advantages of SPH in similar contexts.

Author Response

Thank for your comments and the recommendations for this manuscript. We have made some responses for your comments. Please see the attachment.

Reviewer 2 Report

Comments and Suggestions for Authors

 REVIEWER’S COMMENTS_water-2926962

The manuscript presents a study on employs CPM, discrete fracture network (DFN), equivalent continuum porous medium (ECPM) and hybrid DFN/ECPM to describe four hydrogeological conceptual models of Cholan formation, verifying them against the results of an in-situ trace test. I recommend that the authors address the following issues to improve the quality of the manuscript for publication.

1.     Manuscript Title: The title of the manuscript is too lengthy. The authors need to rephrase the title.

2.     Lines 15-17: The authors should remove the following sentences as it is not necessary: “The continuous porous medium model (CPM) can be applied in the gravel formation, while the Cholan formation, according to the geological investigation, exhibits characteristic of a fracture medium but is often treated as a porous medium since it is a young formation.”

3.     Abstract and Manuscript Title: This study is aimed at employing the use of CPM, discrete fracture network (DFN), equivalent continuum porous medium (ECPM) and hybrid DFN/ECPM to describe four hydrogeological conceptual models of Cholan formation, verifying them against the results of an in-situ trace test. The title of the manuscript does not reflect the aim of the study as indicated in the Abstract section. The authors need to restructure the title of the manuscript to align it with the aim of the study.

4.     Fig. 2: The authors did not state the implication of the result presented in Fig. 2. What resulted in the rise and fall of tracer concentration and what does this rise and fall of tracer concentration connotes?

5.     Lines 339-340: The use of personal pronoun should be avoided. The authors need to remove “We” and restructure the sentence.

6.     Line 353- 354: The authors only presented the results of Fig. 5b to 8b without discussing the implications of the results presented. They only presented the results as follows: “The outcomes of the first screening are shown in Figure 5(b) to Figure 8(b).”. The authors should discuss these results.

7.     Lines 362-364: The authors only presented the results of Fig. 5d to 8d and Fig. 9 without discussing the implications of the results presented. The authors only presented the results of Fig. 5d to 8d and Fig. 9 without discussing the implications of the results presented. They only presented the results as follows: “Figure 5(d) to Figure 8(d) show the number of particles arriving and their travel time for each case. Given that each particle represents a tracer concentration of 208.325 ppb, the results have been converted into concentration-time curves and compared with the in-situ tracer breakthrough curve, shown in Figure 9.

8.     Lines 393-394: The authors should change: “We constructed four hydrogeological conceptual models based on the characteristics of the in-situ geological formation in this study.” to “Four hydrogeological conceptual models based on the characteristics of the in-situ geological formation was constructed in this study.

9.     Conclusion: The conclusion of the study should be presented in one paragraph. The authors need to collapse the three paragraphs of the conclusion to one paragraph.

10.  Conclusion: The authors need to present the implication of the findings of the study towards the end of the conclusion section. They should state what the findings of this study can be used for.

11.  This manuscript deals more on geology and seems not to be suitable for Water journal. I think the manuscript is more suitable for journals on geological formation.

Comments on the Quality of English Language

The quality of English is good except for minor issues.

Author Response

(The authors gave the same response as above.)

Reviewer 3 Report

Comments and Suggestions for Authors

Kindly see the attached file

Comments on the Quality of English Language

Moderate editing of English language required. Authors may kindly request some expert in editing and English to improve their manuscript.

Author Response

(The authors gave the same response as above.)

Round 2

Reviewer 1 Report

Comments and Suggestions for Authors

Despite the authors having correctly pointed out the novelties of the work in their reply, they are still not properly reported in the introduction.

The authors' response regarding FEM and SPH (with appropriate supporting references) must be included either in the introduction or in the conclusion.

Author Response

Thank you for your comments on this article. In response to your comments, we have responded as shown in the attachment.

Reviewer 2 Report

Comments and Suggestions for Authors

The manuscript has been sufficiently improved for publication in Water journal.

Author Response

Thank you for your insightful comments that enrich this article.

Reviewer 3 Report

Comments and Suggestions for Authors

Thanks for incorporating all my comments.

Author Response

(The authors gave the same response as above.)
